# Analysis of Family Structure and Paternity Test of Tan Sheep in Yanchi Area, China

**DOI:** 10.3390/ani12223099

**Published:** 2022-11-10

**Authors:** Ling Li, Yefang Li, Qing Ma, Shuqin Liu, Yuehui Ma, Lin Jiang

**Affiliations:** 1National Germplasm Center of Domestic Animal Resources, Institute of Animal Science, Chinese Academy of Agricultural Sciences (CAAS), Beijing 100193, China; 2College of Animal Science and Technology, Qingdao Agricultural University, Qingdao 266109, China; 3Institute of Animal Science, Ningxia Academy of Agriculture and Forestry Sciences, Yinchuan 750002, China

**Keywords:** Tan sheep, single nucleotide polymorphism, family structure, paternity test, inbreeding coefficient

## Abstract

**Simple Summary:**

This study investigated the family structure, parent–child pair and inbreeding coefficient of a Tan sheep breeding farm in Ningxia, China. The results can be used as the theoretical basis for selecting excellent breeding stock and making an apposite mating plan; furthermore, it can provide an important scientific basis for the conservation and reproduction of Tan sheep germplasm resources.

**Abstract:**

Tan sheep is a special breed of locally protected sheep in China, one of the best quality meat sheep in the world. Due to the unclear pedigree of the rams on the Ningxia Tan sheep breeding farm, we investigated 74 rams in the field and explored a new method for family division. Genomic DNA was extracted from the blood of breeding rams. Using Plink software, GCTA tools and R language, we analyzed the genetic structure, kinship, and inbreeding coefficient of the breeding sheep, which revealed the genetic relationship between the individuals. The results showed that there was no obvious clustering phenomenon in the PCA, and the genetic background of the samples was similar. The G matrix and IBS distance matrix indicated that most individuals were far away from each other. Paternity testing identified 24 pairs of unknown parent–child pairs, and all the Tan sheep could be divided into 12 families, which provided a reference for sheep breeding. The average inbreeding coefficient based on the ROH of this population was 0.049, so there was a low degree of inbreeding and the rams in the field were able to maintain high genetic diversity. Overall, we explored a more accurate method through paternity and kinship analysis; it provides a scientific basis for pedigree construction, which has an important application value for Tan sheep breeding.

## 1. Introduction

As one of the “five treasures” in Ningxia, China, the Tan sheep has tender meat, a very light taste, even fat distribution, rich nutrition, and is the top quality mutton favored by consumers; it is listed as a high-quality sheep breed under national and local protection. Since the 1950s, it has been introduced by more than a dozen other provinces in China, but all of them failed to maintain the original variety characteristics due to unsuitable ecological conditions. Later, due to various factors, the distribution range was significantly reduced; only Yanchi County is listed as a Tan sheep breeding conservation area by the Ningxia Hui Autonomous Region Government and the national Tan sheep breeding farm is also based in Yanchi County. At present, the excellent breeding rams are mainly provided by breeding farms, enterprises, and farmers. However, after long-term, enclosed breeding, decline is likely, due to the consequences of inbreeding.

The wrong pedigree will slow down the breeding process and reduce the population’s genetic diversity and the effective populations in animal breeding [1]. For breeding farms, it will lead to great economic losses. Researchers have simulated ten populations of milking cows with a pedigree error rate of 10%; after 20 years of genetic evaluation, it was found that the estimated breeding value had encountered serious deviation; the genetic gain increased by 4.3% after correcting the pedigree [2]. Therefore, the correct pedigree is significant to breeding farms and seed conservation farms.

Single nucleotide polymorphism (SNP), as the third-generation DNA molecular marker, has the advantages of dense distribution, low typing error, genetic stability and low cost. It is widely used in evolutionary research [3,4,5,6,7,8] and genetic relationship identification [9,10,11,12,13,14]. One study revealed the genetic diversity of the global goat population and highlighted the migratory routes after domestication based on a 50K SNP chip [15]. In addition, SNP technology provides a more convenient technical means for livestock breeding. In this research, we conducted a phylogenetic analysis based on SNP markers, divided the families of the samples and supplemented the unknown relationships between individuals with paternity tests; finally, we evaluated the degree of inbreeding of the population, and analyzed the genetic structure and diversity information of the Tan sheep conservation population from the DNA level, so as to provide a scientific basis and theoretical guidance for the breeding and genetic resource protection of Tan sheep.

## 2. Materials and Methods

### 2.1. Animals

A total of 74 male Tan sheep, with complete ear numbers and birth dates, were selected from the Tan sheep breeding farm in Yanchi, Ningxia. Blood samples of 10 mL were collected from the jugular vein in EDTA anticoagulant tubes, upside down to prevent coagulation, brought back to the laboratory at low temperature, and stored in the refrigerator at −80 ℃ for future use.

### 2.2. Genotyping and Quality Control

Genomic DNA was extracted from 3 mL blood samples using Promega Wizard Genomic Purification according to the manufacturer’s protocol Kits. Nanodrop 1000 (Thermo Fisher Scientific, Waltham, MA, USA) was used to detect the purity and concentration of genomic DNA, and all DNA samples with a ratio of light absorption (A260/280) between 1.8 and 2.0 and a concentration > 50 ng/µL were eligible for genotyping. Individual genotyping was carried out using a 50 K SNP chip (Beijing Compson Biotechnology Co., LTD., Beijing, China).

We used PLINK (V1.90) [16] software to quality control SNP markers. The quality control criteria were: call rates of more than 90%, minor allele frequency (MAF) of more than 0.05, and Hardy–Weinberg equilibrium (HWE) of 1 × 10^−5^. In addition, linkage disequilibrium should be removed when analyzing genetic structure. The window size was set as 1000, the number of moving steps was set as 5, and the r^2^ threshold (multiple correlation coefficient for an SNP being regressed on all other SNPs simultaneously) was set as 0.5, which was used to simplify SNPs.

### 2.3. Principal Component Analysis

Principal component analysis (PCA) transforms multiple linearly correlated variables (SNPs) into a few linearly independent variables with a large explanatory degree of variation through a series of matrix transformations, so as to reveal the genetic background of samples and assist subsequent research. We used PLINK to calculate PCA (--pca 3) and visualized it by the ggplot2 package in Rstudio (V4.1.2) software.

### 2.4. Genetic Relationship and Family Construction Analysis

To calculate the relatedness between pairwise samples, the study used GCTA [17] tool to construct the genomic G matrix, which could more truly reflect the kinship between individuals. The identity by state (IBS) distance matrix by PLINK is based on the genetic information of an organism. NJ (neighbor-joining) tree is a tree-like branch graph to analyze the kinship between various organisms. On the tree, each node represents the near common ancestor of each branch, and the length of line segments between nodes corresponds to the evolutionary distance. The closer the distance is, the closer the kinship is. We used MEGA (V7.0.26) [18] software to generate NJ tree [19], and ITOL online tool (https://itol.embl.de/, accessed on 1 November, 2022) to beautify.

Combining the analysis of genetic relationship and structure, we could roughly judge which Tan sheep were closely related and might come from the same family.

### 2.5. Paternity Test

In order to verify the reliability of the family construction results, considering that paternity testing was used to correct the above grouping, the parent–child relationship was determined by the LOD value of the likelihood method, and the family grouping was determined by combining the aforementioned clustering results. PLINK software was used to specifically screen SNP loci that could be used for paternity testing in this population. The quality control standards were as follows:The individual detection rate and call rate were more than 95%;Minimum allele frequency 0.4;All loci conform to Hardy–Weinberg equilibrium;Linkage disequilibrium was removed, and the spacing between adjacent SNPs on each chromosome was greater than 10 Mb.

The genetic diversity parameters of all loci were calculated by Cervus (V3.0.7) [20] software, including the number of alleles per locus and expected heterozygosity (He) at each locus; polymorphic information content (PIC) and combined exclusion probability (CEP) of the three scenarios. In the simulation analysis, the parameters of the simulated offspring were set as 10,000, the genotyping error rate was 0.01, and the confidence was 80 and 95%. We evaluated the parent–child relationship of individuals according to the likelihood theory, a positive value of LOD indicated that the parent–child relationship was established, that is, the candidate father was the real parent of the offspring. When there were two or more candidate fathers whose LOD value was greater than 0, the one with the higher LOD value was preferred.

### 2.6. Inbreeding Coefficient Analysis

Runs of homozygosity (ROHs), which are generated by the complete transmission of homologous haplotypes from parents to offspring, refer to the continuous segments of homozygous genotypes within an individual and are widely present in all populations [21]. DetectRUNS package was used to calculate the number of the inbreeding coefficient based on ROH $F_{ROH}$, and the formula was as follows:(1)$$F_{ROH}=\frac{\sum L_{ROH}}{L_{auto}}\;$$
where, $L_{ROH}$ is the sum of ROH lengths in individual genomes, and $L_{auto}$ is the total length of sheep autosomal genomes.

## 3. Results

### 3.1. SNP Quality Control Results

A total of 64734 SNPs were detected, and 41149 SNPs were left for subsequent analysis after quality control (Table 1).

### 3.2. PCA Results

PCA visualization results showed (Figure 1) that the ram samples were scattered without obvious clustering. PC1, PC2 and PC3 could respectively explain 3.38, 2.76 and 2.64%, and the vast majority of individuals were distributed in the positive half axis of PC2.

### 3.3. Genetic Relationship and Family Construction Results

We measured the IBS genetic distance ranging from 0.188 to 0.329, with an average distance of 0.297 (Appendix A). In Figure 2, each square represents the genetic distance between pairs of individuals, and the color from blue to red indicates the genetic distance from near to far. The squares on the diagonal represent the genetic distance of individuals themselves. The results showed that most of the individuals were genetically distant from each other, and several individuals in the lower right corner were clustered separately and had a large genetic distance from the other rams.

The relatability value based on the G matrix is shown in Appendix A. In Figure 3, each small square represents the relatability value between the first sample and the last sample. The larger the value, the closer the relatability is. The figure is distributed with four individual aggregations and several dark areas of two to three individual aggregations, which are estranged from the other rams.

Corresponding to the G matrix, the NJ (neighbor-joining) tree showed several branches with 2–3 individuals clustered together, which were, respectively, defined as a family, including family groups 2, 4, 5, 9 and 12. Here, we classified individuals on the same branch as a single family, thus the sample was divided into 14 families (Appendix A). Different groups in the evolutionary tree are marked in different colors (Figure 4).

### 3.4. Paternity Test Results

Through the above screening, 76 SNPs were identified, and He was about 0.50, with an average of 0.5033. Ho ranged from 0.494 to 0.649, with an average Ho of 0.518 and PIC equal to 0.375, and the cumulative exclusion probability of candidate parents for this combination was greater than 99.99% (Table 2). With confidence greater than 95%, 24 pairs of individuals were assigned parent–child relationships, with LOD values between 0.28 and 15.63 and within three mismatch sites (Table 3). All individuals in this test with confidence below 80% were not assigned parent–child relationship. (Appendix A).

The 24 pairs of individuals corresponded to our preliminary family division, and we found that most of the individuals with parent–child relationships were grouped into the same family on the evolutionary tree. In addition, 190351 and 180407 had a parent–child relationship, with an LOD of 2.38 and a confidence of more than 95%, but they were divided into different branches in the evolutionary tree. Therefore, we combined all the branches they were in, this is 12, 13 and 14 in the above Figure 4, which finally resulted in 12 families. The detailed grouping results are shown in Table 4.

### 3.5. Inbreeding Coefficient Results

The mean $F_{ROH}$ was 0.049, and the highest $F_{ROH}$ was 0.199 in 200379. The second was 170291, $F_{ROH}$ was 0.128, which was greater than 0.125 (Appendix A). The lowest inbreeding number was 170071 with $F_{ROH}$ of 0.002. It can be clearly seen from Figure 5 that the proportion of samples corresponded to different inbreeding numbers. Generally speaking, the inbreeding number of samples is low, and the degree of inbreeding is also low, which maintains high genetic diversity.

## 4. Discussion

### 4.1. Family Construction

Pedigree is crucial to animal genetics and breeding research. However, in the breeding process of individual investors and even farms, there will be inevitable mistakes, resulting in incomplete or incorrect pedigree records, particularly in developing countries [22]. Several studies [23,24,25] have paid attention to pedigree construction by genetic relationship and genetic distance between individuals. In fact, paternity tests can also supply part of the pedigree and solve the uncertain parent–child relationship in the population [26,27,28]. Here, we combined the two methods to construct a family, and further added paternity tests to correct the pedigrees of individuals.

The samples we collected were all from the same field, which has been closed breeding over a period of time, with some individuals inevitably having near or far relatives. Hence, no obvious grouping was found and the genetic background was similar in Tan sheep by PCA. The IBS distance matrix and G matrix indicated that some individuals were closely related to each other. Attention should be paid to the mating work between these rams and their female parents in the process of breeding conservation to prevent the risks caused by inbreeding. In addition, there were several individuals clustered separately on the diagonal, which could not be divided into families. Therefore, when constructing the NJ tree, we classified a single and the individual with the closest genetic distance on the tree as the same family, so as to preliminarily determine 14 family groups.

### 4.2. SNP Accuracy

Our criteria for screening SNPs refer to the Simmental cattle paternity test in China [29], except that this trial population was smaller than Zhang’s and used more informative and less numerous markers, so we increased the average marker distance per chromosome to 10 Mb, and the MAF was narrowed to greater than 0.45. Large SNP spacing can effectively avoid linkage disequilibrium between loci, and a MAF close to 0.5 will maximize the strength of parent–child relationship [30]. These conditions enabled us to obtain 76 high-quality SNP locus combinations. The cumulative exclusion probability of a single parent was 99.996% when the genotypes of both parents were unknown, 99.999986% when the single parent genotype was known, and 99.9999998736% when the genotypes of both parents were known. Comparing the parent–child relationship between Simmental cattle and Holstein cattle cross populations, the cumulative exclusion probability of the single and parental inference of 50 efficient SNPs with an average MAF of 0.43 reached 99.797 and 99.9999%, indicating that the 76 markers in this study had high accuracy and credibility. We initially divided the family composition according to genetic distance and relatability and had the birth date of each sample. On this basis, paternity tests would be more effective in distinguishing the relationship between individuals, which is consistent with Fisher’s results [31]. In conclusion, the combination of SNPs in this experiment with the paternity tests carried out on the Tan sheep allowed us to then correct the wrong relationships in the pedigrees.

Then, we detected 24 pairs of paternity through paternity identification, with LOD values ranging from 0.28 to 15.63, with confidence higher than 95%, indicating high accuracy. Corresponding to the evolutionary tree, we found that most parent–child pairs belonged to the same branch, and only one parent–child pair (190351 and 180407) was located in different branches. Interestingly, there was a third family between the two, so we combined them to end up with 12 families. On the one hand, it verified the reliability of the family construction mentioned above; on the other hand, it supplemented some unknown parent–child relationships in the field and explored a new method for family division based on SNP markers, which is of great significance for the future breeding work of Tan sheep, by tapping the potential of breeds and protecting breed resources. Since the samples collected were all rams, there are some limitations in mating with ewes; we will add ewe samples to further clarify the individual relationships of Tan sheep and promote breeding efforts in future.

### 4.3. Degree of Inbreeding

Preventing the increase in inbreeding in livestock and poultry is an important issue to ensure the sustainable development of agricultural and animal husbandry production. The inbreeding coefficient is usually used to evaluate the degree of inbreeding of individuals. By comparing various methods for calculating the inbreeding coefficient, it was found that the ROH-based inbreeding coefficient $F_{ROH}$ was the most accurate and could best reflect the true inbreeding level of individuals [32,33,34,35]. The average $F_{ROH}$ of the samples in this study was 0.049, which showed an upward trend compared with that in 2019 [36]. The reason may be that the inbreeding coefficient is affected by generations; however, breeders of Tan sheep have mainly adopted traditional breeding methods in recent years, and elite rams often undergo a certain degree of closed breeding. Fortunately, the inbreeding of most individuals has remained at a low level. Only 200379 and 170291 had high inbreeding levels ($F_{ROH}$ > 0.125). Technicians in the field should pay special attention to the mating situation of the two to avoid an inbreeding depression. In general, the degree of inbreeding in this population was low and the genetic diversity was high.

## 5. Conclusions

We investigated the genetic background, family structure, parent–child relationship and the level of inbreeding using 50 K sheep chips on a survey of the Yanchi Tan sheep breeding farm in Ningxia, China. A more accurate method was explored through paternity and kinship analysis; it provides a scientific basis for future selection, breeding and pedigree construction. In addition, the degree of inbreeding in this population was low and the genetic diversity was high. This study is of great significance for the genetic evaluation, new breed selection and breed resource conservation of Tan sheep.

## Figures and Tables

**Figure 1 animals-12-03099-f001:**
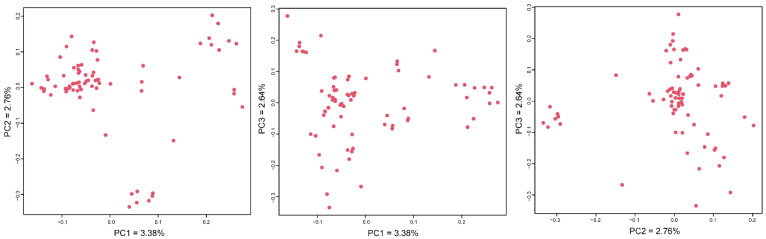
PCA analysis results.

**Figure 2 animals-12-03099-f002:**
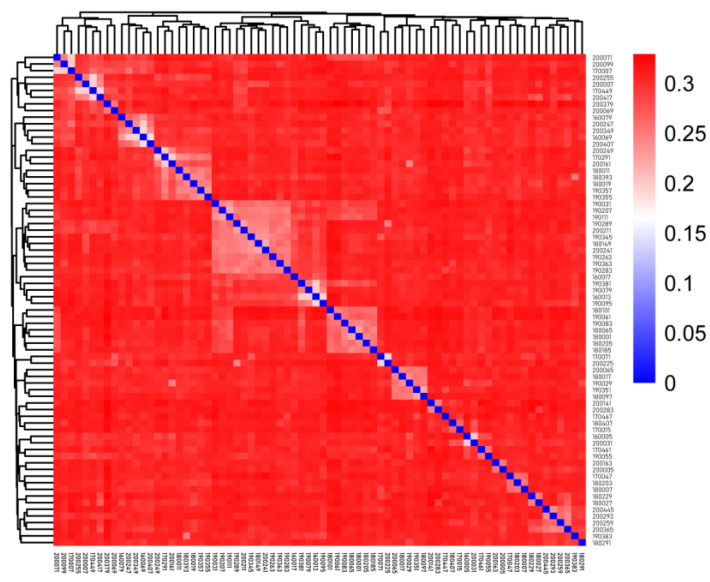
IBS distance matrix visualization in Tan sheep.

**Figure 3 animals-12-03099-f003:**
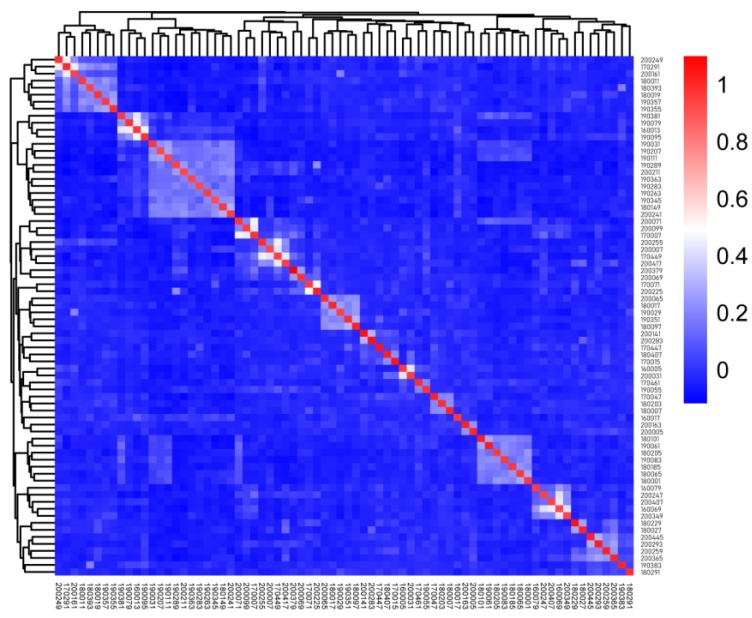
G matrix visualization of this population in Tan sheep.

**Figure 4 animals-12-03099-f004:**
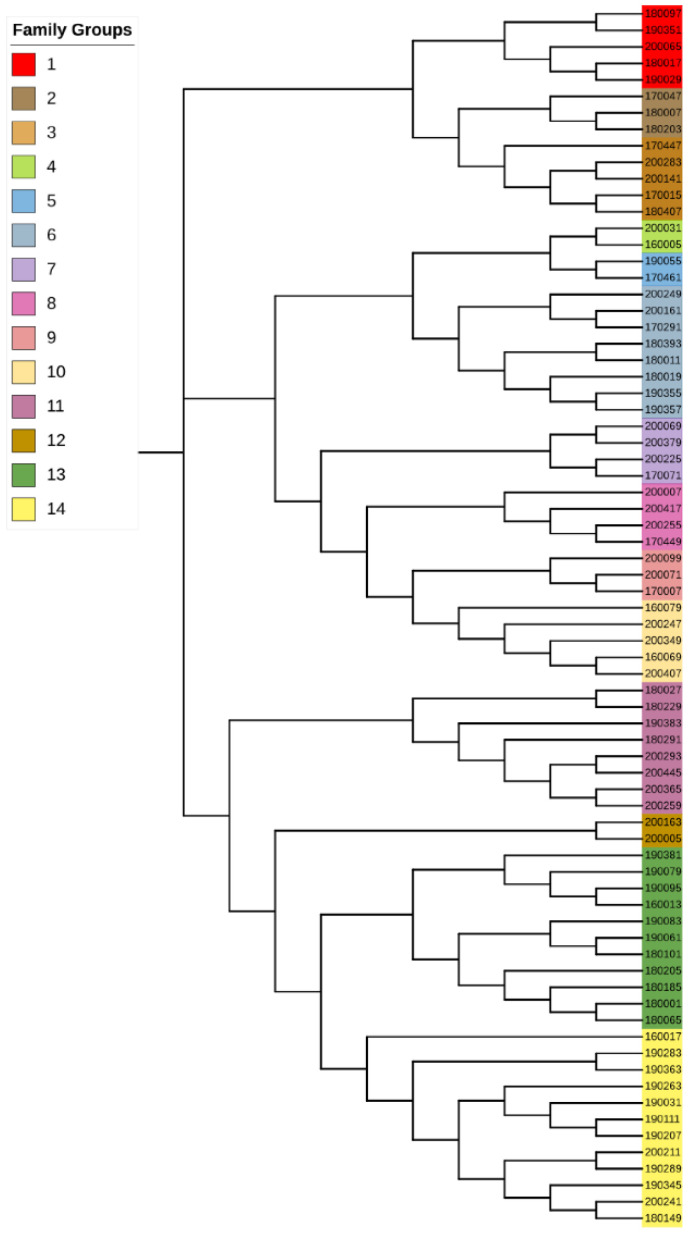
NJ tree in Tan sheep.

**Figure 5 animals-12-03099-f005:**
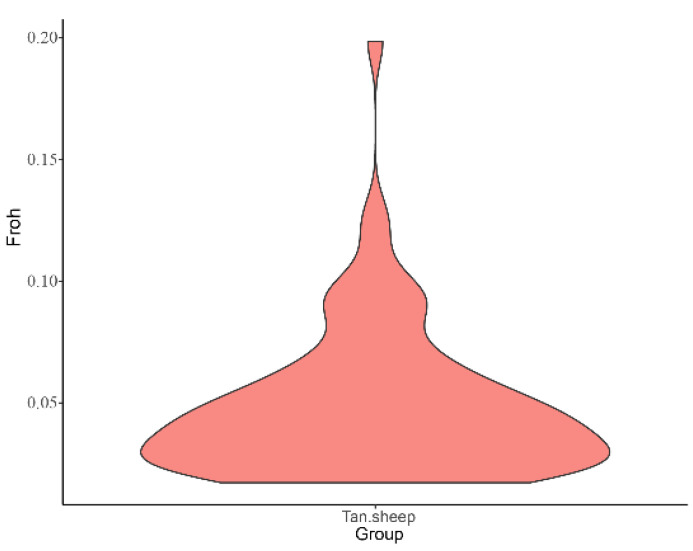
Violin plot of genomic inbreeding in Tan sheep. The width of the graph indicates the probability density distribution of the population $F_{ROH}$. The wider part of the graph indicates the larger number of samples at that level, and the smaller, the number of samples at that level.

**Table 1 animals-12-03099-t001:** Single nucleotide polymorphism chip marker-based quality filtering results.

Parameters	Excluded SNPs	Remaining SNPs
Total number of SNPs before quality control		64734
SNP call rate < 0.90	1472	63262
MAF < 0.05	10162	53100
HWE (*p* < 1 × 10^−5^)	1343	51757
Independent-pairwise 1000 5 0.5	10608	41149
Total number of SNPs after quality control		41149

**Table 2 animals-12-03099-t002:** SNP marker information.

SNP	
Number of loci:	76
Mean number of alleles per locus:	2
Mean expected heterozygosity:	0.5033
Mean polymorphic information content (PIC):	0.3749
Combined non-exclusion probability (first parent):	0.00003936
Combined non-exclusion probability (second parent):	0.00000014
Combined non-exclusion probability (parent pair):	1.264 × 10^−11^

**Table 3 animals-12-03099-t003:** Twenty-four paternity pairs were identified.

Offspring ID	Candidate Father ID	Loci Typed	Pair Loci Mismatching	Pair LOD Score	Pair Confidence
190079	160013	76	0	13.54	*
190357	190355	76	0	12.17	*
190095	160013	76	0	12.15	*
190029	180097	76	3	1.15	*
190351	180407	76	2	2.38	*
190381	160013	76	0	12.9	*
190083	190381	76	2	0.28	*
190355	190357	75	0	12.17	*
180065	180001	76	0	8.11	*
200349	160069	76	0	12.32	*
200225	170071	75	0	13.54	*
200407	160069	76	0	8.37	*
200007	170449	76	0	8.85	*
200099	170007	76	0	9.01	*
200249	170291	76	0	15.63	*
200247	160069	76	0	11.65	*
200031	160005	76	0	8.02	*
200417	170449	76	0	9.67	*
180001	180065	76	0	8.11	*
200241	160017	76	2	0.27	*
190061	190381	76	0	9.51	*
200161	170291	76	0	19.03	*
200255	170449	76	0	8.76	*
200071	170007	76	0	9.56	*

* means paternity assignment with 95% confidence.

**Table 4 animals-12-03099-t004:** Results of family grouping.

Group	Individuals							
Family 1	190029	190351	180097	180407	200283	200141	180017	200065
	170015	170447	170047	180007	180203			
Family 2	170461	190055						
Family 3	200031	160005						
Family 4	190357	190355	170291	200161	200249	180019	180011	180393
Family 5	170071	200225	200379	200069				
Family 6	170449	200255	200417	200007				
Family 7	170007	200071	200099					
Family 8	200407	160069	200349	200247	160079			
Family 9	200445	200293	200259	200365	180291	190383	180229	180027
Family 10	200005	200163						
Family 11	180065	180001	190061	190083	160013	190095	190079	190381
	180185	180205	180101					
Family 12	160017	200241	190345	190289	200211	190207	190111	190031
	190263	190363	190283	180149				

## Data Availability

The data presented in this study are available on request from the first author/corresponding authors.

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
