# Peer review of "Analysis of Family Structure and Paternity Test of Tan Sheep in Yanchi Area, China"

_animals, 2022, doi:10.3390/ani12223099_

Round 1
Reviewer 1 Report
In this work the authors aimed to uncover the family structure of 74 rams belonging to the Tan Sheep breeds located in the Yanchi Area of China. The authors wanted to gain knowledge to reduce the potential inbreeding by adopting the correct breeding strategies for a breeds very important in the Yanchi County. Different methodolody were used to assess the pedigree by genetic relashipship and family construction analyses.
The work does not want to be ambitious but very simple in the design however it add knowledge usefull to the farmers for to adopt the correct breeding in the breed.
Minor modifications are needed to improve the intelligibility of the work:
Line 58: change the upper format
Line 73: specify the Promega kit used
Line 77: specify the bead chip used
Line 81: put -5 in uppercase format
Line 99: include the link to the ITOL software
Line159: because figure 3 is not clearly visible, the comment in lines 159-161 is not clear. Erase the words “the closer is at line 158 because are a repetition
Line 168: not clear statement: Family contained within individual numbers between 2-12 so please elucidate it.
Line 200: figure 5. Not clear the explanation of the violin plot. Not visible the white dot, the edges, the black box
Line 210: change the format of the reference
Author Response
Thank you for your valuable advice. We immediately revised it in strict accordance with your requirements, and checked the statement of the paper on this basis. Please kindly accept the detailed information in the attachment.

Reviewer 2 Report
Manuscript is entitled "Analysis of Family Structure and Paternity Test of Tan Sheep in Yanchi Area, China". This study divided Tan sheep families using a novel and creative method. The author corrected the pedigree through paternity test and supplemented the unknown parental information, which is of great significance to the conservation and utilization of Tan sheep in Yanchi area. The research content is rich and meaningful. However, some existing defects should be addressed before publication. The main problems are as follows:
1. The abstract doesn’t reflect the review of the "new method" of family division. Please supplement and explain it clearly.
2. The cluster line at the top of Figure 3 is not clear, please modify it.
3. Line 67: "Tan male sheep" should be replaced by " male Tan sheep".
4. Line 80, change "、" to " , ".
5. Line 81, change "1×10-5" to "1×10-5".
6. Line 90, note the use of brackets.
7. The Figure 4 might be used more obvious colors to distinguish different families.
8. Why did you choose 76 SNPs? Would it be better to reduce some SNPs?
9. In general, there is not in-depth enough discussion. It is suggested to compare the results carefully with other researchers.
10. Therefore, we combined the branches of the NJ tree in which these two families were located, resulting in 12 families. Please express this sentence clearly, including which pedigrees are being combined.
11. It is noted that your manuscript needs careful editing by someone with expertise in technical English editing paying particular attention to English grammar, spelling, and sentence structure so that the goals and results of the study are clear to the reader.
12. Table 3. 24 paternity pairs were identified. Please keep this format consistent with other tables.
13. The samples in this paper were only studied on rams, there are certain limitations in the selection and breeding of male ewes. We recommend that authors include ewes information if available.
Author Response

(The authors gave the same response as above.)

Reviewer 3 Report
This manuscript proposes a pipeline to construct family structure of 74 Chinese Tan sheep. It is rather usefully when the pedigree is absent or not clear, and can be used for other farm animals. However, I have some questions and concerns regarding the incomplete description of the methods.
Major comments:
In “introduction” section, the author says wrong pedigree will lead to biased EBV. However, to what extend the biased EBV can be modified after performing this analysis? Or how this analysis can reduce the error rate of pedigree?
What’s the accuracy of Paternity Test? Have you used the information of age in Paternity Test?
It seems that the sections of “Data Availability Statement” and “Acknowledgments” were just copied from the template without any modification.
Minor comments:
Line 80: Normally, MAF means minor allele frequency. Please make sure of it.
Line 83: The definition of “R2” should be given.
Line 90: The R package for visualization should be given.
Line 99: How you generated the NJ tree? Normally, MEGA was used to generate NJ tree and ITOL was used to draw the tree. Besides, the data and the parameters used in tree construction should be given.
Line129: “the number of inbreeding lines” is not clear to me.
Line141: How you calculate the variance of the first three PC? It would be better to offer the formula.
Line167: I am not clear about how these samples were divided into 14 families. At least the threshold should be given.
Author Response

(The authors gave the same response as above.)
